# Polydopamine-Coated Laponite Nanoplatforms for Photoacoustic Imaging-Guided Chemo-Phototherapy of Breast Cancer

**DOI:** 10.3390/nano11020394

**Published:** 2021-02-04

**Authors:** Renna Liu, Fanli Xu, Lu Wang, Mengxue Liu, Xueyan Cao, Xiangyang Shi, Rui Guo

**Affiliations:** 1College of Chemistry, Chemical Engineering and Biotechnology, Donghua University, Shanghai 201620, China; renna_liu906@163.com (R.L.); xufanli1992@163.com (F.X.); wanglu2013@dhu.edu.cn (L.W.); 13262579590@163.com (M.L.); caoxy_116@dhu.edu.cn (X.C.); 2State Key Laboratory for Modification of Chemical Fibers and Polymer Materials, College of Materials Science and Engineering, Donghua University, Shanghai 201620, China

**Keywords:** polydopamine, laponite, chemotherapy, phototherapy, photoacoustic imaging

## Abstract

Theranostic nanoplatforms combining photosensitizers and anticancer drugs have aroused wide interest due to the real-time photoacoustic (PA) imaging capability and improved therapeutic efficacy by the synergistic effect of chemotherapy and phototherapy. In this study, polydopamine (PDA) coated laponite (LAP) nanoplatforms were synthesized to efficiently load indocyanine green (ICG) and doxorubicin (DOX), and modified with polyethylene glycol-arginine-glycine-aspartic acid (PEG-RGD) for PA imaging-guided chemo-phototherapy of cancer cells overexpressing α_v_β_3_ integrin. The formed ICG/LAP-PDA-PEG-RGD/DOX nanoplatforms showed significantly higher photothermal conversion efficiency than ICG solution and excellent PA imaging capability, and could release DOX in a pH-sensitive and NIR laser-triggered way, which is highly desirable feature in precision chemotherapy. In addition, the ICG/LAP-PDA-PEG-RGD/DOX nanoplatforms could be uptake by cancer cells overexpressing α_v_β_3_ integrin with high specificity, and thus serve as a targeted contrast agent for in vivo PA imaging of cancer. In vivo experiments with 4T1 tumor-bearing mouse model demonstrated that ICG/LAP-PDA-PEG-RGD/DOX nanoplatforms exhibited much stronger therapeutic effect and higher survival rate than monotherapy due to the synergetic chemo-phototherapy under NIR laser irradiation. Therefore, the reported ICG/LAP-PDA-PEG-RGD/DOX represents a promising theranostic nanoplatform for high effectiveness PA imaging-guided chemo-phototherapy of cancer cells overexpressing α_v_β_3_ integrin.

## 1. Introduction

Phototherapy is an emerging light-triggered cancer treatment, including photothermal therapy (PTT) to convert light into heat and photodynamic therapy (PDT) to generate highly reactive oxygen species (ROS) for the ablation of cancer [1,2,3,4]. With the aid of photosensitizers, solid tumors could be effectively treated with high specificity and reduced side effects by controlling the irradiation site, and can even be diagnosed by photoacoustic (PA) imaging with high spatial resolution and good tissue contrast [5,6]. However, phototherapy could not address the issues of tumor relapse, which may cause the high death rate. In contrast, chemotherapy could exhibit good therapeutic effect by systemic administration, but face the problems, such as lack of tumor selectivity, severe side-effect, and poor release property. Therefore, integrating both photosensitizers and anticancer drugs into one nanocarrier has become one of the strategies to overcome the limitation of single-mode therapy, and construct tumor-targeted theranostic nanoplatform for PA imaging-guided chemo-phototherapy [7].

In the past few years, various kinds of nanoplatforms for combinational tumor chemo-phototherapy have been developed [8,9,10,11,12,13,14,15,16]. For example, poly (dopamine) nanoparticles (PDA) could be utilized as photothermal agents and load anticancer drugs doxorubicin (DOX) or 7-ethyl-10-hydroxy-camptothecin on surface via π-π stacking interaction, which showed synergetic effect for in vitro and in vivo cancer treatment [17]. In Wu et al.′s study, mesoporous silica coated PDA functionalized reduced graphene oxide (rGO) nanosheets were synthesized to load DOX and modify hyaluronic acid (HA) on surface [18]. Under laser irradiation, the photothermal effect of rGO and PDA may weaken the interaction between DOX and nanoplatform, leading to the increased drug release. The obtained nanoplatforms exhibited targeted chemo-photothermal therapy against cancers overexpressing CD44. Xu et al. constructed dual targeting HA and arginine-glycine-aspartate (RGD) conjugated silica-coated gold nanorods to load DOX, and demonstrated that their better therapeutic effect than single chemo or photothermal therapy [19]. It is encouraging that nanoplatforms uniting photothermal therapy and chemotherapy showed great advantages and synergetic effect in the treatment of tumor, but few studies focus on developing theranostic nanoplatforms by utilizing suitable photosensitizers for the simultaneous diagnosis of tumor and monitoring the distribution of nanoplatforms via PA imaging.

As an FDA-approved photosensitizer, indocyanine green (ICG) has attracted wide interest due to its photothermal and photodynamic therapeutic characteristics [20], because it can efficiently convert the absorbed NIR laser energy into heat [21] and produce ROS as well [22]. Duan et al. employed monolayered-double-hydroxide (MLDH) to load ICG and DOX, and demonstrated their excellent tri-modal synergetic anticancer activity [23]. In Liu et al.’s study, PEG modified graphene oxide quantum dots were developed to load DOX and ICG, which displayed much stronger tumor growth inhibition than monotherapy [24]. Moreover, ICG could be applied in the detection of tumor by PA imaging under NIR laser with the aid of nanocarriers to improve its stability in vivo [25,26]. For instance, Shen et al. built ICG-loaded and folic acid-modified multiwalled carbon nanotubes, and demonstrated that the nanoplatforms exhibited excellent photostability for tumor targeted PA imaging and phototherapy [27]. He et al. merged metal organic framework with hollow mesoporous organosilicon nanoparticles (NPs) by PDA interlayer to encapsulate ICG and DOX for cancer theranostics [28]. Chen et al. synthesized hollow copper sulfide NPs to co-load DOX and ICG, and camouflaged cancer cell membrane on its surface for homologous targeting PA imaging and chemo-phototherapy [29]. As a result, a high-performance PA-imaging guided chemo-phototherapy theranostic nanoplatform needs to load both ICG and anticancer drugs efficiently, provide high chemical/colloidal stability, deliver therapeutic agents to tumor specifically and release in a stimuli-responsive/controlled manner.

Laponite (LAP) is a type of synthetic nanoclay with excellent colloidal stability, biocompatibility and biodegradability. Due to its unique layered structure, LAP has been widely used as nanocarriers to load drugs or stabilize imaging agents for tumor treatment and diagnosis [30,31,32,33,34]. In our previous study, ICG was loaded in the interlayer of laponite (LAP) by a surprisingly high efficiency (94.1%), and PDA was then coated on the ICG-LAP hybrid surface with PEG-RGD conjugated as targeting agents [35]. The presence of PDA and LAP improved the stability of ICG dramatically, and imparted the formed ICG/LAP-PDA-PEG-RGD NPs with enhanced photothermal and photodynamic therapeutic effect, rendering them perfect starting point nanoplatform for further combination with chemotherapy and PA imaging. In this study, anticancer drug DOX was further loaded onto ICG/LAP-PDA-PEG-RGD nanoparticles, and the formed ICG/LAP-PDA-PEG-RGD/DOX nanoplatform was characterized by various techniques to verify the PA imaging capability and drug release property under laser irradiation and different pH environments. In vitro and in vivo experiments were carried out to assess their RGD-mediated targeted delivery, PA imaging and synergistic chemo-phototherapeutic effect under laser irradiation using 4T1 cells (a mouse breast cancer cell line) overexpressing integrin α_v_β_3_ as a model.

## 2. Materials and Methods

### 2.1. Synthesis and Characterization of ICG/LAP-PDA-PEG-RGD/DOX NPs

LAP nanodisks were dispersed in acetate buffer (9 mg/mL, pH = 5.0), and ICG solution (2 mg/mL) was added under magnetic stirring for 4 h. After centrifugation and rinsing, the formed ICG/LAP were collected and dispersed in a mixture of ethanol, dopamine (DA) and ammonia aqueous solution for the self-polymerization of DA to form ICG/LAP-PDA NPs. Then NH_2_-PEG-RGD solution was added in the purified ICG/LAP-PDA solution in tris buffer (pH = 9.0) under magnetic stirring for 24 h. Targeted ICG/LAP-PDA-PEG-RGD and non-targeted ICG/LAP-PDA-*m*PEG were obtained after purifying and lyophilized to preserve in dark [35]. Finally, the ICG/LAP-PDA-PEG-RGD was dispersed in phosphate buffer solution (5.8 mg, pH = 8.0), and mixed with doxorubicin (DOX) aqueous solution (1.4 mg) under magnetic stirring for 6 h. Then ICG/LAP-PDA-PEG-RGD/DOX were purified by ultrafiltration centrifuge tube (Millipore, Billerica, MA, USA) with a MWCO of 10,000 (8000 r/min, 5 min) for 3 times to remove the free DOX.

The structure, stability and PA property of ICG/LAP-PDA-PEG-RGD/DOX were characterized by different techniques, and the drug release properties with/without laser irradiation and at different pH were also studied (details in Appendix A).

### 2.2. Cytotoxicity and Cellular Uptake of ICG/LAP-PDA-PEG-RGD/DOX NPs

Cytotoxicity, cellular uptake and chemo-phototherapeutic effect of ICG/LAP-PDA-PEG-RGD/DOX were evaluated (details in Appendix A).

### 2.3. In Vivo PA Imaging and Chemo-Phototherapy of ICG/LAP-PDA-PEG-RGD/DOX NPs

In Vivo PA imaging and chemo-phototherapeutic effect of ICG/LAP-PDA-PEG-RGD/DOX were evaluated on a subcutaneous 4T1 tumor-bearing mouse model (Details in Appendix A).

## 3. Results and Discussion

### 3.1. Synthesis and Characterization of ICG/LAP-PDA-PEG-RGD/DOX

In this work, photosensitizer ICG was firstly trapped within the interlayer of LAP nanodisks efficiently [35], and PDA were formed on the surface of LAP by self-polymerize of dopamine under alkaline condition to provide additional protection and photothermal capability, followed by conjugating PEG-RGD as targeting moiety to form ICG/LAP-PDA-PEG-RGD NPs. In addition, then anticancer drug DOX was loaded on surface by π–π stacking interactions with PDA to construct the ICG/LAP-PDA-PEG-RGD/DOX nanoplatforms, and their targeted chemo-phototherapy and PA imaging capability to cancer cells overexpressing α_v_β_3_ integrin were investigated (Scheme 1).

Firstly, UV-vis spectra of ICG/LAP-PDA-PEG-RGD/DOX and ICG/LAP-PDA-PEG-RGD were measured (Figure 1a). The emerging absorption at 480 nm in the spectrum of ICG/LAP-PDA-PEG-RGD/DOX should be ascribed to the DOX loaded on ICG/LAP-PDA-PEG-RGD by π–π stacking interaction between DOX and PDA shell [17]. In addition, the loading efficiency was calculated to be as high as 87.6%. SEM image and size distribution results in Figure 1b further demonstrated that ICG/LAP-PDA-PEG-RGD/DOX displayed uniform spherical shapes with an average diameter of 64.6 nm, which is consistent with our previous work [35]. Then DLS was applied to evaluate the hydrodynamic diameter and Zeta potential of ICG/LAP-PDA-PEG-RGD/DOX (Appendix A). It is demonstrated that drug-loaded nanoplatforms showed no significant change in hydrodynamic diameter of 148.2 ± 3.1 nm and a slight decrease of surface potential to −17.10 ± 0.95 mV. In addition, the hydrodynamic size kept almost constant in water, PBS and culture medium DMEM over 7 days (Appendix A), indicating their good colloidal stability. This should be attributed to the PEG chain modified on the outer layer of ICG/LAP-PDA-PEG-RGD/DOX, which might prolong their circulation time and increase their accumulation at tumor.

The drug release property of ICG/LAP-PDA-PEG-RGD/DOX was evaluated at pH 5.0 and 7.4 to emulate the tumor site and physiological environment with/without laser irradiation (Figure 1c). Under pH 7.4, only less than 7% of DOX was released in 24 h, while DOX released under pH 5.0 was over 33%. The pH-sensitive release property should be attributed to the hydrophilic form of DOX with protonated amine groups at weak acid environment in comparison to its hydrophobic form at neutral environment [32]. The less release of DOX during blood circulation under physiological pH environment may reduce the in vivo side effect of nanoplatforms, and much higher amount of DOX released in tumor site at pH 5.0 could effectively exert the anticancer effect. More interestingly, a burst release of drug could be observed under laser irradiation. The release rate under pH 5.0 was almost doubled to 61% at 24 h, while only about 17% of DOX was released at physiological environment. The NIR-triggered release property could be attributed to the diminished interaction between DOX and PDA by photothermal heating [18], which may accelerate the drug release and improve chemotherapeutic effect at tumor site under laser irradiation. Thus, the ICG/LAP-PDA-PEG-RGD/DOX nanoplatforms were successfully synthesized with a high drug loading efficiency, and would specifically release drug at tumor sites in a pH-sensitive and NIR-triggered manner, which is highly desirable feature in precision tumor chemotherapy.

### 3.2. Photothermal and Photoacoustic Property of ICG/LAP-PDA-PEG-RGD

The photothermal conversion abilities of ICG/LAP-PDA-*m*PEG and ICG/LAP-PDA-PEG-RGD solutions were investigated by irradiation of 808 nm laser for 3 min (Figure 2a). Different from the constant temperature of LAP and water after irradiation, the temperature of ICG increased to 39.7 °C, while ICG/LAP-PDA-*m*PEG and ICG/LAP-PDA-PEG-RGD were heated up to about 45.3 °C. Their better photothermal conversion ability than free ICG solution could be explained by the improved stability of ICG and the PDA surface [35]. Then their photoacoustic imaging ability were investigated by measuring the PA imaging under laser irradiation (Figure 2b). Obviously, both non-targeted ICG/LAP-PDA-*m*PEG and targeted ICG/LAP-PDA-PEG-RGD solutions showed a bright PA signal, and their PA intensity enhanced gradually with the increase of ICG concentration. In summary, ICG/LAP-PDA-PEG-RGD NPs have a high photothermal capability for phototherapy and could be an effective contrast agent for PA imaging.

### 3.3. Cytotoxicity Assay

Firstly, the viability of mouse breast cancer 4T1 cells treated with ICG/LAP-PDA-PEG-RGD for 24 h was measured by CCK-8 assay to evaluate their biocompatibility. As shown in Figure 3a, over 85% cells kept alive in the studied concentration range, demonstrating that ICG/LAP-PDA-PEG-RGD NPs are non-cytotoxic to cells. Then α_v_β_3_-positive and -negative tumor cells (4T1 and MCF-7) were used to determine the targeting capability of ICG/LAP-PDA-PEG-RGD to cancer cells overexpressing α_v_β_3_ integrin by the cellular uptake experiment [36]. After 6 h of incubation with ICG/LAP-PDA-*m*PEG and ICG/LAP-PDA-PEG-RGD, the cellular Si concentration were determined by ICP-OES, and RGD-block group was also set as control by incubating cells with 5 µM RGD before the adding of NPs. It is obvious that for MCF-7 cells-expressed lower α_v_β_3_ integrins (Figure 3b), although the cellular Si uptake enhanced with the increase of NP concentration, there is no significant difference among three groups. In contrast, for 4T1 cells (Figure 3c), ICG/LAP-PDA-PEG-RGD group exhibited a significantly higher Si uptake than both ICG/LAP-PDA-mPEG and RGD-block group at the same ICG concentration. The over twofold increase of Si cellular uptake demonstrated that ICG/LAP-PDA-PEG-RGD could be specifically and effectively uptake by 4T1 cells overexpressing α_v_β_3_ integrin via the RGD-mediated targeting effect.

Finally, the chemo-phototherapy effect of ICG/LAP-PDA-PEG-RGD/DOX nanoplatforms was assessed by measuring the viability of 4T1 cells treated with ICG/LAP-PDA-PEG-RGD/DOX with/without the 808 nm laser irradiation for 5 min (Figure 3d). It is obvious that PBS group showed no notable cytotoxicity before and after laser irradiation. For DOX group, the cell viability decreased notably in a concentration-dependent way, and laser irradiation seems to have no notable improvement on the therapeutic effect. In contrast, although ICG/LAP-PDA-PEG-RGD displayed almost no inhibition effect at all concentrations, they showed a significant reduction in cell viability after laser irradiation (*p* < 0.001). This should be ascribed to the photothermal and photodynamic therapy effect of loaded ICG and coated PDA, which is in agreement with our previous study [35]. For drug-loaded ICG/LAP-PDA-PEG-RGD/DOX, the cell viability reduced gradually with the growth of DOX concentration, and their lower therapeutic effect than free DOX should be ascribed to the low effective DOX concentration on cells due to the slow drug release [37]. However, under laser irradiation, ICG/LAP-PDA-PEG-RGD/DOX could exhibit a significant lower cell viability than ICG/LAP-PDA-PEG-RGD (*p* < 0.001) and free DOX group (*p* < 0.001), indicating the synergistic effect of phototherapy and chemotherapy. After calculation, the half-maximal inhibitory concentration (IC50) of ICG/LAP-PDA-PEG-RGD/DOX + L (1.55 µg/mL) was much lower than that of ICG/LAP-PDA-PEG-RGD + L (1.73 µg/mL) and DOX + L (4.11 µg/mL). This result illustrated the combination of laser irradiation and ICG/LAP-PDA-PEG-RGD/DOX could reach an extraordinary inhibition effect at a relatively low drug concentration, which may improve therapeutic effect and alleviate the side effect of anticancer drugs. Therefore, with NIR laser irradiation, ICG/LAP-PDA-PEG-RGD/DOX could effectively exert chemotherapy and phototherapy for cancer treatment.

### 3.4. In Vivo Photoacoustic Imaging

To evaluate their in vivo photoacoustic imaging property, targeted ICG/LAP-PDA-PEG-RGD and non-targeted ICG/LAP-PDA-*m*PEG solution were intravenously injected in 4T1 tumor bearing mice (Figure 4a,b). Both nanoparticles showed an increased PA signal at tumor sites after injection, and the PA signal peaked at 2 h post injection, likely due to the accumulation of NPs at tumor. More interestingly, ICG/LAP-PDA-PEG-RGD group displayed a 2.4 times stronger signal than ICG/LAP-PDA-*m*PEG at 2 h (*p* < 0.001), and the gap between them became even larger at 4 h. This result demonstrated that the specific recognition between overexpressed α_v_β_3_ integrin on 4T1 cells and ICG/LAP-PDA-PEG-RGD could increase their active accumulation at tumor and prolong their retention. It is worth noting that free ICG solution could not present a clear image of tumor due to its rapid degradation and non-specific binding in vivo [38]. Then, the biodistribution of both targeted and non-targeted nanoparticles were assessed by ICP-OES via analyzing Si content in tumor and major organs at 2 h. Similar to the PA imaging result, the Si concertation of ICG/LAP-PDA-PEG-RGD at tumor was more than double that of ICG/LAP-PDA-*m*PEG (*p* < 0.001), and the accumulation of NPs at other organs was relatively low, indicating that ICG/LAP-PDA-PEG-RGD could specifically accumulate at tumor overexpressing integrin α_v_β_3_. Overall, ICG/LAP-PDA-PEG-RGD nanoplatforms have the potential as an efficient PA imaging contrast agent.

### 3.5. In Vivo Evaluation of Photo-Chemotherapy

In Vivo chemo-phototherapeutic effect of ICG/LAP-PDA-PEG-RGD/DOX was measured by the subcutaneous 4T1 tumor-bearing mouse model. For laser irradiation groups, an 808 nm laser (2.5 cm^2^) was irradiated on tumor for 5 min after intratumor injection of PBS, ICG/LAP-PDA-PEG-RGD and ICG/LAP-PDA-PEG-RGD/DOX on Day 0 and 3. Both the digital photographs and average tumor volume in Figure 5a,b showed that laser irradiation did not alter the tumor growth of PBS control group. Without laser irradiation, ICG/LAP-PDA-PEG-RGD/DOX group exhibited an inhibition effect within 12 days, and then a rapid increase of tumor size occurred probably because of the gradual metabolism of drug-loaded nanoplatforms. This result indicated that single chemotherapy could not efficiently inhibit the growth of tumor. For single phototherapy, laser irradiation at 808 nm caused a remarkable tumor growth inhibition in ICG/LAP-PDA-PEG-RGD group. More importantly, ICG/LAP-PDA-PEG-RGD/DOX displayed an even better antitumor effect with a significant decrease of tumor size than single chemotherapy and phototherapy group (*p* < 0.001), verifying the synergistic effect of chemo-phototherapy combination. The body weights increased steadily during the experiments, implying the neglectable side-effect of nanoplatforms, but the survival rates changed greatly between different groups. On day 30, all mice in PBS with/without laser and ICG/LAP-PDA-PEG-RGD group died, and only 20% mice in ICG/LAP-PDA-PEG-RGD/DOX group were alive, indicating the limited therapeutic effect of single chemotherapy. In contrast, with laser irradiation, both ICG/LAP-PDA-PEG-RGD and ICG/LAP-PDA-PEG-RGD/DOX group kept 100% survival rate until 36 days after treatment, and then mice began to die in single phototherapy group. These results clearly verified that ICG/LAP-PDA-PEG-RGD/DOX could display significantly stronger tumor growth inhibition than monotherapy due to the synergetic anticancer activity of phototherapy and chemotherapy.

Finally, the in vivo therapeutic effect of ICG/LAP-PDA-PEG-RGD/DOX was verified by Hematoxylin-eosin (H&E) staining and TUNEL staining of tumor sections after treatment (Figure 6). Obviously, there was no sign of necrosis in both PBS group, indicating the 808 nm laser alone cannot kill cancer cells. ICG/LAP-PDA-PEG-RGD showed a damage of cell nuclei only under the irradiation of an 808 nm laser, illustrating their phototherapeutic effect. It is worth mentioning that more extensive necrosis and dramatic changes in cell morphology were observed for ICG/LAP-PDA-PEG-RGD/DOX group probably because of the combination of chemo-phototherapy. In addition, TUNEL staining displayed a similar trend of necrosis in different treatments as yellow signals (Figure 6b). The cell apoptosis rates of ICG/LAP-PDA-PEG-RGD/DOX, ICG/LAP-PDA-PEG-RGD+L, ICG/LAP-PDA-PEG-RGD/DOX+L were calculated to be 36.5%, 51.2%, and 86.5%, respectively, revealing that the therapeutic efficiency of ICG/LAP-PDA-PEG-RGD/DOX is highly improved by the combination of chemotherapy and phototherapy. Finally, H&E staining of major organs from mice after ICG/LAP-PDA-PEG-RGD/DOX treatment were measured in Appendix A. Nanoplatform group showed no obvious pathological damage in major organs in comparison with PBS group, illustrating their good biocompatibility in vivo. Taken together, the formed ICG/LAP-PDA-PEG-RGD/DOX could be an effective and targeted theranostic nanoplatform for PA-guided chemo-phototherapy.

## 4. Conclusions

In conclusion, we developed a polydopamine-coated LAP nanoplatform for PA imaging-guided chemo-phototherapy of integrin α_v_β_3_-overexpressed cancers. Photosensitizer ICG and anticancer drug DOX were efficiently loaded in LAP nanodisks and on PDA shell separately, and PEG-RGD were modified on surface as targeting agents. The formed ICG/LAP-PDA-PEG-RGD/DOX nanoplatforms displayed good colloidal stability, excellent photothermal and PA imaging properties, and could release DOX in a pH-sensitive and NIR-triggered way. In addition, the ICG/LAP-PDA-PEG-RGD nanoplatforms could be specifically and efficiently uptake by 4T1 cells with integrin α_v_β_3_ overexpression, and serve as a targeted contrast agent for in vivo PA imaging of 4T1 cancer. Finally, in vivo experiments demonstrated that ICG/LAP-PDA-PEG-RGD/DOX nanoplatforms could exhibit significantly stronger therapeutic effect and higher survival rate than monotherapy by the synergistic chemo-phototherapy under NIR laser irradiation. Therefore, the formed hybrid ICG/LAP-PDA-PEG-RGD/DOX could be an effective nanoplatform for PA imaging-guided chemo-phototherapy of cancer cells overexpressing α_v_β_3_ integrin, which may shed light on future innovation of nanoplatforms for precise cancer theranostics.

## Data Availability

All data generated or analyzed during this study are included in this published article (and its Appendix A).

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
