# Peer review of "Polydopamine-Coated Laponite Nanoplatforms for Photoacoustic Imaging-Guided Chemo-Phototherapy of Breast Cancer"

_nanomaterials, 2021, doi:10.3390/nano11020394_

Round 1
Reviewer 1 Report
This work presents polydopamine (PDA)-coated laponite (LAP) nanoparticles that were loaded with ICG and doxorubicin as welll as PEG-RGD. The results are impressive and comprehensive. Overall, some suggestions for improvement are provided:
1. Some background on ICG for PTT should be provided including gold nanoparticle (Huang et al., Adv. Res, 2010, 1, 13–2), graphene (Yang et al., Nano Lett, 2010, 10, 3318–3323) and liposome approaches (e.g. Miranda et al., Biomater Sci . 2019; 7-3158).
2. Some background on clinical elements of PTT for cancer should be provided (Xi et al., Nat Rev Clinical Oncol, 17, 657, 2020) as well as chemophototherapy (Luo et al., Adv Sci. 2017; 4, 1600106).
3) Fig 1A) Where is ICG absorption? It is unusual if it not present for a photoacoustic contrast agent, which requires strong NIR absorptino?
Author Response
Response to Reviewer 1 Comments:
This work presents polydopamine (PDA)-coated laponite (LAP) nanoparticles that were loaded with ICG and doxorubicin as welll as PEG-RGD. The results are impressive and comprehensive. Overall, some suggestions for improvement are provided:
- Based Some background on ICG for PTT should be provided including gold nanoparticle (Huang et al., Adv. Res, 2010, 1, 13–2), graphene (Yang et al., Nano Lett, 2010, 10, 3318–3323) and liposome approaches (e. g. Miranda et al., Biomater Sci. 2019; 7-3158).
Author reply: We thank the reviewer for his/her comments. According to the reviewer’s suggestion, we have added corresponding literatures based on ICG for PTT as reference 14-16 (e.g., Huang et al., Adv. Res, 2010, 1, 13-2, Yang et al., Nano Lett, 2010, 10, 3318-3323 and Miranda et al., Biomater Sci. 2019; 7-3158).
- Some background on clinical elements of PTT for cancer should be provided (Xi et al., Nat Rev Clinical Oncol, 17, 657, 2020) as well as chemophototherapy (Luo et al., Adv Sci. 2017; 4, 1600106).
Author reply: According to the reviewer’s suggestion, we have provided corresponding literatures based on clinical elements of PTT for cancer as reference 3 and 4 (e.g., Xi et al., Nat Rev Clinical Oncol, 17, 657, 2020 and Luo et al., Adv Sci. 2017; 4, 1600106).
- Fig 1A) Where is ICG absorption? It is unusual if it not present for a photoacoustic contrast agent, which requires strong NIR absorption?
Author reply: According to our previous study (Nanomaterials, 2018, 8, 347), the strong absorption peak of ICG in NIR region disappeared after the coating of PDA on ICG/LAP. This result is different from the appearance of ICG absorption after loading ICG on surface of PDA in other studies.

Reviewer 2 Report
The authors have done a good piece of work on developing a doxorubicin and indocyanine green loaded nanoparticle for chemo-phototherapy application on Breast Cancer cells.
Although there are certain aspects which the authors have missed out addressing.
- The developed nanoparticle lacks the basic characterization: The size of the nanoparticle is measured by DLS but the actual size is not determined i.e. electron microscopy (TEM/ SEM). DLS gives the hydrodynamic diameter which could be misleading as this nanoparticles have so many functional modifications and loading agents, the actual size is vital for a nanoparticle for its application if it is considered for use in the pre-clinical trials or any future applications.
- For a cell biology study, it is always essential to have two cell lines to compare the targeting efficacy. In this case, I would recommend to use a αvβ3 negative cell line along side 4T1. Although the authors have showed an experiment blocking the receptor with RGD peptide before treating with the nanoparticle but having a positive and negative cell lines gives a more convincing result.
- The in-vivo study lacks the information on toxicity. The authors have showed the Si concentration on other organs and also measured body weight during the study but there is no data which validates that there is no sign of toxicity in other organs. It would be ideal that the authors could do a H&E staining of other organs tissues and check for signs of toxicity after targeted and non-targeted delivery or with and without irradiation.
Author Response
Reply to Reviewer 2 Comments:
The authors have done a good piece of work on developing a doxorubicin and indocyanine green loaded nanoparticle for chemo-phototherapy application on Breast Cancer cells.
- The developed nanoparticle lacks the basic characterization: The size of the nanoparticle is measured by DLS but the actual size is not determined i.e. electron microscopy (TEM/SEM). DLS gives the hydrodynamic diameter which could be misleading as this nanoparticle have so many functional modifications and loading agents, the actual size is vital for a nanoparticle for its application if it is considered for use in the pre-clinical trials or any future applications.
Author reply: We thank the reviewer for his/her comments. According to the reviewer’s suggestion, we measured the scanning electron microscope (SEM) image and size distribution of the ICG/LAP-PDA-PEG-RGD/DOX in Figure 1b in the revised manuscript to confirm the size of nanoparticle. The result has been added and discussed in the revised manuscript on Line 138-140 on Page 4. See also below:
“SEM image and size distribution results in Figure 1b further demonstrated that ICG/LAP-PDA-PEG-RGD/DOX displayed uniform spherical shapes with an average diameter of 64.6 nm, which is consistent with our previous work.”
- For a cell biology study, it is always essential to have two cell lines to compare the targeting efficacy. In this case, I would recommend to use a αvβ3 negative cell line along side 4T1. Although the authors have showed an experiment blocking the receptor with RGD peptide before treating with the nanoparticle but having a positive and negative cell lines gives a more convincing result.
Author reply: According to the reviewer’s suggestion, we have added the targeting efficacy of ICG/LAP-PDA-PEG-RGD by using a αvβ3 negative cell line (MCF-7) in Figure 3b in the revised manuscript. These results have been added and discussed in the revised manuscript on Line 192-195 and Line 198-200 on Page 5. See also below:
“Then αvβ3-positive and -negative tumor cells (4T1 and MCF-7) were used to determine the targeting capability of ICG/LAP-PDA-PEG-RGD to cancer cells overexpressing αvβ3 integrin by the cellular uptake experiment.”
“It is obvious that for MCF-7 cells-expressed lower αvβ3 integrins (Figure 3b), although the cellular Si uptake enhanced with the increase of NP concentration, there is no significant difference among three groups.”
- The in-vivo study lacks the information on toxicity. The authors have showed the Si concentration on other organs and also measured body weight during the study but there is no data which validates that there is no sign of toxicity in other organs. It would be ideal that the authors could do a H&E staining of other organs tissues and check for signs of toxicity after targeted and non-targeted delivery or with and without irradiation.
Author reply: Thanks for the reviewer’s suggestion. We have added the H&E staining of major organs from mice after ICG/LAP-PDA-PEG-RGD/DOX treatment as described on Page 5 Line 8-11 in Supporting Information. The result was shown in Figure S2 and discussed in the revised manuscript on Page 8 Line 300-303. See also below:
“Finally, H&E staining of major organs from mice after ICG/LAP-PDA-PEG-RGD/DOX treatment were measured in Figure S2. Nanoplatform group showed no obvious pathological damage in major organs in comparison with PBS group, illustrating their good biocompatibility in vivo.”

Round 2
Reviewer 2 Report
The authors have succesfully addressed the commnents and completed the missing links in the manuscripts. It can be accepeted in the present form.